# An Assessment of Reprocessed GPS/MET Observations Spanning 1995-1997

Anthony J. Mannucci[1], Chi Ao[1], Byron A. Iijima[1], Thomas K. Meehan[1], Panagiotis Vergados[1], E. Robert Kursinski[2], William S. Schreiner[3]

[1]Jet Propulsion Laboratory, California Institute of Technology, Pasadena, CA, 91109, USA
[2]PlanetiQ, Golden, CO, 80401, USA
[3]University Corporation for Atmospheric Research, Boulder, CO, 80307, USA

*Correspondence to*: Anthony J. Mannucci (anthony.j.mannucci@jpl.nasa.gov)

**Abstract.** We have performed an analysis of reprocessed GPS/MET data spanning 1995-1997 generated by the COSMIC Data Analysis and Archive Center (CDAAC) in 2007. CDAAC developed modified dual-frequency processing methods for the encrypted data (anti-spoofing (AS)-on) during 1995-1997. We compared the CDAAC data set to the Modern-Era Retrospective Analysis for Research and Applications-2 (MERRA-2) reanalysis, separately for AS-on and AS-off, focusing on the altitude range 10-30 km. MERRA-2 did not assimilate GPS/MET data in the period 1995-1997. To gain insight into the CDAAC data set, we developed a single-frequency data set for GPS/MET, which is unaffected by the presence of encryption. We find excellent agreement between the more limited single frequency data set and the CDAAC data set: the bias between these two data sets is consistently less than 0.25% in refractivity, whether or not AS is on. Given the different techniques applied between the CDAAC and the new data set presented here (designated JPL), agreement suggests that the CDAAC AS-on processing and the single frequency processing are not biased in an aggregate sense greater than 0.25% in refractivity, which corresponds approximately to a temperature bias less than 0.5 K. Since the profiles contained in the new single frequency data set are not a subset of the CDAAC profiles, the combination of the CDAAC data set, consisting of 9,579 profiles, and the new single-frequency data set, consisting of 4,729 profiles, yields a total number of 11,531 unique profiles from combining the JPL and CDAAC data sets. All numbers are after quality control has been applied by the respective processing activities.

# 1 Introduction

There is currently intense interest in Earth science observations that are useful for measuring decadal scale changes to the climate (Wielicki et al., 2013). When considering such observations from orbiting platforms, combining observations from multiple missions must be addressed. Therefore, measurement accuracy needs to be characterized over time scales that exceed the lifetime of any single mission (Gleisner et al., 2020).

The Global Navigation Satellite System (GNSS) radio occultation (RO) technique has the potential to provide such long-term observations with accuracies that are sufficient to meet the stringent demands of climate change observation, documented as 0.05 K temperature stability per decade (Leroy et al., 2006; Ho et al., 2009; Steiner et al., 2013; GCOS, 2016). GNSS RO has been used in climate trend studies that use climate model finger-printing techniques (Lackner et al., 2011) or the observations directly (Steiner et al., 2009; Steiner et al., 2020a; Vergados et al., 2021). A significant challenge in observing decadal-scale climate change is that geophysical variables display significant variation on time scales shorter than tens of years. Such "natural variability" is potentially the most significant factor hindering the interpretation of observations relevant to climate change (Leroy et al., 2008; Santer et al., 2017). It is widely recognized that the longer a time series is, the more valuable it is for observing climate change (Leroy et al., 2008; Wielicki et al., 2013).

In this paper, we analyze observations from the RO record dating back to 1995 when the first observations were obtained from the "proof-of-concept" GPS/MET mission (Kursinski et al., 1996; Rocken et al., 1997). GPS/MET data have been used successfully in climate-related studies, but generally using only a limited subset of the acquired data consisting of two three-week periods when the data quality was highest (Oct 1995 and Feb 1997) (Steiner et al., 2009; Steiner et al., 2011). There exists a much more extensive data set from 2007 that includes many more months in the period 1995-1997 that has not yet been evaluated in the literature but that could contribute to more robust trend detection.

The two limited GPS/MET periods coincide with two favorable conditions that contributed to increased data quality: 1) lack of encryption of the second transmitted GPS frequency (L2, 1227.60 MHz), thus permitting nominal signal-to-noise ratios (SNR) for processing and 2) favorable orientation of the spacecraft permitting the aft-directed antenna to be aligned with the velocity direction of the spacecraft (so-called "prime" periods; Schreiner et al., 1998). Signal encryption is also known as "Anti-Spoofing" (AS). In this paper, we distinguish periods when encryption was active, also called "AS-on", from periods when encryption was temporarily disabled ("AS-off"). Prime periods, which had the favorable antenna orientation, could occur when encryption was on or off. Throughout the GPS/MET mission, one or the other or both of these conditions periodically occurred.

Past climate studies have tended to use data only from the periods when both the favorable antenna orientation and AS-off were present. Such periods permit standard dual-frequency (DF) processing of the observations, which is used to reduce the ionospheric impact on the retrievals (Rocken et al., 1997; Hajj et al., 2002). There are two published studies using data with an encrypted second frequency and thus reduced signal-to-noise ratios (SNRs) (Nishida et al., 2000; Randel et al. 2003). Randel et al. (2003) noted no discernable difference comparing periods with and without encryption. However, these studies did not use the data to analyze trends, and they did not use the reprocessed data set from the University Corporation for Atmospheric Research (UCAR) COSMIC Data Analysis and Archive Center (CDAAC) published in 2007. We note that all studies to date (including this one) use data only from "prime" periods of favorable spacecraft orientation.

The main distinguishing feature of this study versus past studies using GPS/MET is a detailed comparison with a reanalysis clearly distinguishing periods when anti-spoofing encryption was turned on versus off. The AS-on period resulted in significantly reduced L2 SNRs that required additional temporal smoothing to be applied to this frequency. This necessitated modified processing techniques that were finalized for the 2007 data set but remained a concern for users of the data interested in climate trends or for evaluation (Steiner et al., 2009; Lackner et al., 2011; Marquardt et al., 2001; Gorbunov and Kornblueh, 2001). UCAR's CDAAC generated such a re-processed data set in 2007 encompassing 9,579 profiles spanning 1995-1997 that has yet to be fully characterized for scientific use. JPL also produced a dual-frequency data set during this period, although it will not be evaluated for this paper.

Recent interest in the potential of using reanalyses for climate trend studies (Bosilovich et al., 2015) motivate interest in these RO data sets from GPS/MET because they extend over the period which reanalyses can be evaluated, and may provide insight into how a uniformly processed reanalysis might be subject to biases that change over time, particularly going back to the 1990s when fewer space-based observations were available than currently. Assessing the structural uncertainty of RO measurements is a major activity (Ho et al., 2009; Ho et al., 2012; Steiner et al., 2013; Steiner et al., 2020b) reflecting that, despite an origin of common raw data, the processing chains of different centers produce different results due to reasonable but different processing choices and algorithms adopted by those centers.

It is the purpose of this work to provide insights on the magnitudes of possible biases of the CDAAC re-processed data set when encryption is present by comparing it with a new data set produced at JPL using single frequency processing. As a corollary benefit, an expanded data set is now available from the period based on these new reprocessed data. As is typical for RO data sets (Ho et al., 2012; Steiner et al., 2020b), different processing strategies that use different methods of quality control do not reject the same set of profiles, possibly indicating the quality control criteria are too strict. The single frequency processing strategy described in this paper is not susceptible to limitations of encryption that are applied to the L2 frequency during periods of AS-on. In this work, we compare the CDAAC dual-frequency data set with new JPL single frequency retrievals during conditions when AS is on or off. We also compare to the MERRA-2 reanalysis (Basiliovich et al., 2015), a

reanalysis spanning 1980 to the present, produced by NASA's Global Modeling and Assimilation Office. MERRA-2 does not assimilate RO data from 1995-1997, so it can be considered an independent source of information about potential biases in GPS/MET data. However, MERRA-2 may itself be biased. An additional "third vote" data set, which is available from single-frequency (SF) processing, provides clarity on what potential biases might be. This is similar in spirit to the three-corner hat technique applied by Anthes and Rieckh (2018) which is used to characterize noise in four data sets.

In this work, we present an analysis of two reprocessed GPS/MET data sets that have so far not been assessed in the literature to our knowledge. Section 2 describes the data sets available for analysis, including a description of how they were obtained compared to standard dual-frequency processing. In Section 3 we present the results of an assessment that compares the two data sets to the MERRA-2 reanalysis product and to each other (CDAAC and JPL). In Section 4 we discuss the results and what they may imply about biases in the GPS/MET record and in the reanalysis. We conclude in Section 5 and provide suggestions for further research.

## 2 Data Sets

GPS/MET data were collected over each month spanning 1995-1997 (Rocken at al., 1997). Assessing the data from the full GPS/MET data set is valuable as a means of assessing reanalyses during the period 1995-1997 or as a prelude to assimilating these data in future reanalysis activities. Reanalysis biases in the upper troposphere/lower stratosphere are shown to be reduced when radio occultation data are assimilated (Poli et al., 2010; Ruston and Healy, 2020).

A reprocessing was performed at CDAAC in 2007. Figure 1 shows statistics from the data set, including global coverage and the number of profiles by month currently available in the CDAAC archive. All data shown are after quality control has been applied. To our knowledge, an analysis of the GPS/MET data reprocessed in 2007 has not been published. The reprocessing used a similar software base to what was used for the COSMIC data set (Schreiner et al., 1998). COSMIC launched in April, 2006 and underwent a first reprocessing in 2007.

The new data set introduced in this paper is a re-processed data set from JPL using a single-frequency processing approach applied to the occulting link from which the atmospheric refractivity information is obtained. A benefit of the single-frequency data set is to assess the impact of AS on the CDAAC dual-frequency data set, since AS encryption of the second GPS frequency required a non-standard processing method to be adopted for the L2 frequency. Single frequency processing does not use the L2 frequency, so potential biases introduced by the non-standard L2 processing may be revealed by comparing the two data sets. Results of these comparisons are discussed in Section 3.

Despite its lower number of profiles, the single frequency data set presented here is not a subset of the CDAAC data set, so combining the CDAAC data set with this new data set will increase the overall number of available profiles from GPS/MET. There is also a dual-frequency data set from JPL that covers both AS-on and AS-off periods. We will not be assessing that data set in this paper. We emphasize that the CDAAC and JPL refractivity data sets presented in this paper are independently processed starting with the raw data, including independent orbit determination for GPS/MET and clock calibrations.

## 2.1 Single-Frequency Processing

It has been recognized since the earliest development of GNSS radio occultation that signal delays caused by the Earth's ionosphere must be calibrated to achieve sufficiently accurate results in the neutral atmosphere (Kursinski et al., 1996). This calibration is usually achieved using the two frequencies that are broadcast by GNSS systems specifically to minimize the influence of the ionized upper atmosphere on precise positioning applications. However, it has been widely reported that dual-frequency calibration methods do not fully remove ionospheric effects (Vorob'ev and Krasil'nikova, 1994; Kursinski et al., 1997). Significant effort has been expended to understand the impact of residual ionospheric biases introduced in the upper troposphere and lower stratosphere (Syndergaard, 2000; Steiner and Kirchengast, 2005; Mannucci, et al., 2011; Vergados et al., 2011; Danzer et al., 2013; Danzer et al., 2020; Liu at al., 2020). Single-frequency processing is subject to different residual effects, primarily due to multi-path on the pseudorange signal (discussed below), so comparison between single and dual-frequency processing is another means of assessing residual biases. In general, and particularly for the more recent generation of GNSS receivers, single frequency processing is expected to have lower precision than dual-frequency processing due to the use of the much noisier pseudorange. However, for climate studies, the precision is less important than biases that may appear between the daytime and nighttime profiles due to ionospheric effects.

The single frequency retrieval processing used in this paper is described next. The foundation of the method is to use the difference between the phase path and pseudorange path of the transmitted signal, where the latter travels at the group velocity of the electromagnetic wave (Davies, 1990). The path changes caused by the dispersive ionospheric medium are of equal magnitude but opposite sign between phase and pseudorange (Davies, 1990). Single frequency radio occultation processing has been described previously in two publications (de la Torre et al., 2004; Larsen et al, 2005). Following a notation similar to that used in de la Torre et al. (2004), we can write the additional phase $L_k$ caused by the neutral and ionized atmospheres as:

$$L_k = \eta - \frac{I}{f_k^2} \qquad (1)$$

where $\eta$ is the (non-dispersive) additional phase path caused by the neutral atmosphere, in excess of the geometric path, and $I/f_k^2$ is the excess phase path caused by the ionosphere, proportional to the column density of electrons along the transmitter-receiver raypath (total electron content). The index $k$ specifies the transmission frequency $f_k$ (=1575.42 MHz for $k$=1 and 1227.60 MHz for $k$=2). The additional path of the pseudorange signal $P_k$ caused by the neutral and ionized atmospheres is:

$$P_k = \eta + \frac{I}{f_k^2} \qquad (2)$$

Due to the much longer wavelength of the ranging code compared to the phase (300 m versus 19 cm for phase) the range precision is much less than the phase precision. The pseudorange is generally not used in radio occultation. However, if only a single frequency is available, a combination of the phase and pseudorange yields a biased estimate of the ionospheric factor $I$, that can then be used to correct changes in the additional phase path at the available $f_1$ frequency (also called L1) to recover changes in the atmospheric delay $\eta$ during the occultation using Equation (1). The range and phase combination is given by:

$$I_1 = \frac{I}{f_1^2} = 0.5\,(P_1 - L_1) \qquad (3)$$

The occultation retrieval proceeds as usual after removing the ionospheric term $I_1$ calculated as above from the phase at the L1 frequency. We term this approach "direct phase correction" because the phase is adjusted before deriving bending angles. This differs from the standard dual-frequency approach that applies the dual-frequency correction to bending angles rather than to the signal phase (Syndergaard, 2000). The standard ionospheric correction in bending angle space, at common impact parameter, is meant to compensate for the raypath separation that occurs between the two frequencies L1 and L2.

As discussed in de la Torre et al. (2004), there are two raypaths for which an ionospheric estimate is required: the occulting link, and the non-occulting link used for calibrating the receiver clock. Since the non-occulting link is affected by the ionosphere only above satellite altitude (~730 km), its impact is much less than on the occulting link which propagates through the maximum region of the electron density profile. Correspondingly, the phase change of the non-occulting link over the occultation time of ~120 seconds is much less for the non-occulting link as the ionosphere tends to vary more smoothly at altitudes above the satellite. Therefore, the single-frequency correction was only applied to the occulting link in this work. The calibration links used two frequencies, despite that the L2 frequency was much weaker, because smoothing can be applied effectively when the ionosphere does not vary rapidly. During the GPS/MET experiment, additional calibration links are required to calibrate the GPS clocks, since they were intentionally dithered using a technique called "selective availability" (see Hajj et al., 2002 for a description of the calibration process). As with the calibration link, the ionospheric contribution for these ground-based links varies negligibly during the occultations, and the L2 frequency can be smoothed without incurring significant error. The software GIPSY/OASIS is used for orbit and clock determination, using processing similar to that described in Hajj et al. (2004). CDAAC processing uses the BERNESE software for orbit and clock determination as described in Schreiner et al. (2009). RO retrieval processing for CDAAC's GPS/MET data is described in Kuo et al. (2004).

Deriving the ionospheric correction from Equation (3) requires several considerations. First, the pseudorange estimate $P_1$ is available at 1-second intervals, whereas the phase data $L_1$ are produced at a 50 Hz rate. Therefore, a means is required to derive the phase change due to the ionosphere at the L1 frequency with a cadence of 50 Hz. Second, the random errors in the pseudorange data are at least a factor of 100 larger than in the phase data. To produce retrieval profiles with similar random uncertainties to the dual-frequency technique requires that temporal smoothing be applied to $I_1$, which is available at the 1 Hz pseudorange cadence. The smoothing algorithm to reduce the pseudorange noise is critical. As has been shown by von Engeln et al. (2009), smoothing algorithms themselves can introduce biases in the data that depend on the level of noise, so care is required to minimize smoothing-related biases for climate applications. After smoothing is applied to $I_1$, it is interpolated to a 50 Hz rate as described below.

The theoretical basis for smoothing the ionospheric contribution to the phase is that the ionospheric contribution varies slowly compared to the noise introduced by the pseudorange. We assume that the slow ionospheric variation is described by a series of low-order polynomials over the altitude range of the occultation, although we do not know in advance what polynomial change in phase is caused by the ionosphere. We therefore use standard least squares methods to estimate the underlying polynomial. This requires that we assume a polynomial order for the fit (linear, quadratic, etc.). The assumption that the ionospheric contribution to phase is slowly varying is valid in the absence of small-scale ionospheric irregularities (Hajj et al., 2002) that cause phase scintillation. In the presence of irregularities, other means are required to assess the potential bias such irregularities may impose (Verkhoglyadova et al., 2015). We do not further consider the potential impact of ionospheric irregularities on the ionospheric correction.

When considering a polynomial-based temporal smoothing algorithm applied to the time series $I_1$, it is clearly advantageous to fit the polynomial over as limited a time range as possible, taking advantage of Taylor's theorem asserting that smooth functions (in this case, the phase change due to ionosphere versus time) are well approximated by low-order polynomials over sufficiently limited time intervals. It is also advantageous to use low-order polynomials to avoid fitting to the noise. However, for a given polynomial fit order, lesser time intervals for the fitting lead to less smoothing and less filtering of the noise. On the other hand, fitting over too long a time interval will introduce biases if the underlying ionospheric variation over time is not well described by the low order polynomial used in the fit.

The approach used by de la Torre et al. (2004) for the occulting link, of which a variant is used here, is to perform linear fits to the single frequency ionospheric estimate from Equation (3) over overlapping moving time windows of 30-seconds duration. The time window is moved forward in time every 1-second range point. The smoothed ionospheric value is recovered from the center of each fit, moving forward in time, except at the edges, where the most central point possible is used. Fitting the linear polynomial over a limited portion of the radio occultation period reduces biases associated with choosing a single low-order polynomial for the entire occultation interval.

The variant to this sliding time window approach that we use here is to combine estimates from multiple fitting sub-intervals,

rather than use only the estimate from one sub-interval. Combining multiple fits reduces the variance of the smoothed result, however it combines quantities with correlated noise in a weighted average. Such an average is not guaranteed to be an unbiased estimator of the true average unless the weights accurately reflect the ratios of the variances of the individual fits, which is generally not the case here (see Section 7.2 of Mandel, 1964 for a derivation of weights that yield an unbiased estimate). Thus, a bias might be introduced in the weighted averaging, even as the variance is reduced. However, a bias

introduced in this way will not be the same from profile to profile and is unlikely to result in a climatological bias when multiple profiles are averaged.

The smoothing approach is illustrated in Figure 2. In the upper part of the figure, the "fitting step" is described. The time domain of the occultation is partitioned into multiple overlapping time regions, starting at 15 km impact parameter and

extending 30 km upwards to 45 km altitude, typically corresponding to a time interval of approximately 9-10 seconds duration. (The lowest altitude for the smoothing is approximately the same lowest altitude used in the standard dual frequency correction. Below 15 km, the ionospheric correction is linearly extrapolated based on the lowest few data points.) Successive time intervals overlap by 90%. A quadratic polynomial is performed during each time interval, fitting the ionospheric estimate according to equation (3) versus time, using the L1-frequency range and phase values available at 1-second cadence. Figure 2 shows a linear

fit example for simplicity of notation, but quadratic fits were actually used. A comparison of linear and quadratic fitting when dual-frequency data were available for GPS/MET (108 occultations on June 23, 1995) showed that linear fitting produced a discernable bias between the single-frequency and standard dual-frequency processing, and the absence of a bias when quadratic fitting was used. A week of processing COSMIC data in January 2008 showed similar results, leading to the selection of quadratic fitting in this study. The order of polynomial fit might need to be revisited in follow-on studies using single-

frequency processing nearer to solar maximum conditions.

The step to evaluate the fits is described in the bottom part of Figure 2. After fits are performed for each time interval, the final smoothed value at a given time is based on a weighted sum of the fits evaluated for each time sub-interval. The weighting function is a simple linear "hat function" as diagrammed in the lower right. Weighting is maximum in the center of the sub-

interval, decreasing linearly to zero at the end of each sub-interval. Weights are normalized such that the sum of all weights equals one. This variation of the weighting for different positions within the sub-interval is chosen because the statistical variance of polynomial fits are smallest near the center of the sub-interval and largest at the ends of the sub-interval. The values of the fits at the ends of the sub-intervals are given the least weight in the weighted sum. The dependence of statistical variance on the position within a subinterval is based on standard propagation-of-error considerations (see Chapter 7 of Bevington and

Robinson, 1992).

Examples of the ionospheric estimates based on this smoothing approach are shown in Figure 3. Plotted in each panel (a-f) are: the ionospheric estimate using phase minus range, the ionospheric estimate using dual-frequency phase (L1-L2), and the fits to these quantities. The original data are at 1-second time intervals, but the fits are evaluated every 20 msec (50 Hz) to create ionospheric-free data for use in upstream processing because the occultation data are sampled at 50 Hz. The altitude range for the fits in 15-60 km. Some occultations do not fully reach 60 km altitude, in which case the highest altitude reached is used as the highest altitude for the smoothing. In Section 3, we only analyze the results up to 30 km altitude, but occultation data above this altitude is used in the occultation processing, so upper altitude ionospheric estimates are needed. The vertical scales in each panel deliberately differ so that detail can be seen in the variation of ionospheric phase contribution across the occultation.

Panels (a) and (b) of Figure 3 are examples where AS is on and the ionospheric estimate from the L1-L2 phase data differ strongly from the single frequency estimate. These two examples did not pass quality control (QC) using dual-frequency processing but did pass QC for single frequency processing. These examples are similar to the AS-on case presented in de la Torre et al. (2004, their Figure 1(b)) in that the variation using L1 alone is larger than using two frequencies. In addition, the dual-frequency data shows a "scalloping" behaviour – linear variation followed by discontinuous jumps in phase. This may indicate that the receiver phase lock loop did not usefully track the phase of the weaker L2 frequency. Panels (c) through (f) (AS-off) show qualitatively that the two estimates of ionospheric variation are similar between dual- and single-frequency techniques. The occultations for panels (c) through (f) passed QC for single frequency processing.

In the following section, we discuss results obtained by comparing these single-frequency processed radio occultations with the dual-frequency re-processed GPS/MET dataset at CDAAC. We show evidence that the median values of single-frequency profiles are not biased with respect to the dual-frequency data, suggesting that both data sets are well calibrated despite using non-conventional ionospheric processing. In the case of CDAAC, additional smoothing is applied to the dual-frequency ionospheric estimate derived from the phase data, to mitigate the larger variance of the L2 data when AS is on. We then compare both data sets to the MERRA-2 reanalysis, which did not assimilate GPS/MET data. Comparing these three data sets suggests that reanalysis biases may exist near the tropical tropopause region during the period 1995-1997.

## 2.2 Data Quantity and Characteristics

The reprocessed CDAAC GPS/MET data set consists of 9,579 profiles spanning the period April 1995-February 1997, distributed as 5,002 with AS-off and 4,577 with AS-on. The temporal distribution of the data is shown in Figure 1. As mentioned earlier, past works that used GPS/MET data for climate or process studied tended to focus only on the two months October 1995 and February 1997. The CDAAC profiles are divided between periods of AS-on and off and so are shown separately. The JPL profiles are shown combined, since the retrieval processing does not distinguish these two conditions. The JPL single-frequency processing has yielded 4,729 profiles after QC, from an original 6,173 before a final QC was applied

based on comparisons to ECMWF analysis and other factors (e.g. see Ho et al., 2009 for a description of quality control criteria). These sets are not completely overlapping: as is typical for RO from different processing centers, different criteria yield different sets of profiles that pass QC. There are 11,531 unique profiles in this combined set. This compares with less that 3,000 CDAAC AS-off profiles contained in the periods Feb 1995/Oct 1997 that were used in previously published studies. We note that JPL used both frequencies where possible for orbit determination and for calibration of the occulting link, where smoothing of the weak second frequency could be applied. Single-frequency techniques were only applied to the occultation retrieval.

The geographic and solar local time distributions of these data sets are also shown in Figure 1. Gaps in the geographic distribution are related to the distribution of ground sites used to support GPS/MET occultations, which were undertaken when Selective Availability was in effect. Selective Availability was an intentional dithering of the GPS clock signals to reduce accuracy for receivers not specifically authorized by the US government. To remove these GPS clock errors, at least two ground sites in common view of the occulting and calibration satellites were required (Hajj et al., 2002).

## 3 Results

The objective of this section is to provide insight into the climate quality of GPS/MET data sets that have not received wide attention in the literature. As described below, our assessment suggests that dual-frequency processing when the L2 signal is weaker (AS-on) is not inherently less reliable than processing during the more limited periods when AS was off. We focus our analysis on the altitude range 10-30 km where the closed-loop tracking techniques implemented by the GPS/MET receiver were effective. We do not offer any conclusions about altitudes outside this range.

Clearly issues do arise during AS-on periods as evidenced by Figure 3 panels (a) and (b) (see also de la Torre et al., 2004). These appear to be cases when the receiver phase-locked-loop could not track the signal. However, such cases do not pass the subsequent QC step and are thus removed from consideration.

We address whether the profiles obtained during the AS-on periods are biased even though close enough to the ECMWF reanalysis to pass QC. To perform our assessment, we have compared profiles from the CDAAC data set that are common with profiles in the new JPL single-frequency data set. If modified dual-frequency processing leads to a bias with AS-on, then it is likely that these profiles will be biased with respect to the single frequency data set, since these two data sets are based on fundamentally different ways of removing ionospheric doppler shifts. The L1 frequency, which is the only frequency used in the JPL approach, is not affected by AS-on or off.

Comparisons between the CDAAC and JPL data sets, for the altitude range 10-30 km, are shown in Figure 4. This figure compares the common set of RO acquisitions that have passed QC in both data sets (2,777 profiles), which we refer to as the "profile-to-profile" (P2P) matched data set (Ho et al., 2012). These are the common set of profiles that result after independent

processing and independent QC at both UCAR and JPL. Throughout the prescribed altitude range, refractivities do not deviate by more than 0.25%, which roughly corresponds to temperature retrieval agreement in the range ~0.5 K (Kuo et al., 2004; Appendix C). The CDAAC AS-on and AS-off retrievals are not appreciably different in comparison to the JPL retrievals. AS status is not distinguished for the JPL data set since only the L1 frequency is used in the retrievals, and L1 is not affected by AS-on or off.


The agreement between the JPL and CDAAC data sets suggests that biases in each data set are limited to at most 0.25%. However, there are other possibilities. The two data sets could share a common bias that would not be revealed in this comparison. If true, this common bias would likely not be due to the presence of AS-on because the JPL processing is insensitive to that condition. Such a bias is not the subject of this work and needs to be determined by other means. Other

climate-related work that has used the GPS/MET data for various purposes, and using processing that precedes the CDAAC 2007 reprocessing, suggests that GPS/MET biases unrelated to AS are not significant (Steiner et al., 2009; Lackner et al., 2011; Randel et al., 2003). It is unlikely that the CDAAC GPS/MET reprocessing would introduce new biases that match biases from the JPL processing. Another possibility is that subsets are biased in opposite signs such that the full data set is unbiased. For example, the GPS/MET retrievals (JPL or CDAAC) could have a negative bias in the southern hemisphere and a similar but

opposite bias in the northern hemisphere that cancels. We will return to this subject later.

It is constructive to compare the CDAAC and JPL data sets to the MERRA-2 reanalysis, which did not assimilate GPS/MET RO data during 1995-1997. We assume that it is unlikely that biases that may exist in the RO data sets would match possible MERRA-2 biases. This comparison, shown in Figure 5, differs from Figure 4, indicating that the biases with respect to

MERRA-2 exceed those found between JPL and CDAAC. For example, at several altitudes the differences exceed the value of 0.25% which is a value that bounds the JPL-CDAAC comparison. Focusing on biases that exceed 0.25% in Figure 5, we note the following features of these differences:

JPL and CDAAC AS-off versus MERRA-2 agree up to an altitude of 22 km, where they start to diverge from each other

significantly. JPL and CDAAC AS-on show a noticeable bias at 15 km altitude, a somewhat lower but noticeable bias at 17 and 19 km, and no significant bias at altitudes above 23 km where there is a more significant bias between JPL and AS-off. Both JPL and CDAAC AS-on have significant biases with respect to MERRA-2 starting at 23 km and continuing upward.

In summary, disagreement between the RO measurements and MERRA-2 exceed the differences between JPL and CDAAC

(Figure 4), and do so in a structured way. Biases between RO and MERRA-2 increase at 15 and 17 km, and above 22 km.

Interpretation of Figure 5 is not straightforward. If the biases exceeding 0.25% were due solely to biases in MERRA-2, then one would not expect different biases between the three data sets plotted in Figure 5. On the other hand, if biases exceeding 0.25% existed in the JPL or CDAAC data sets, it is difficult to explain the small differences between the JPL and CDAAC

data sets (Figure 4), unless that bias were common to both JPL and CDAAC. We have already discussed why such a common bias is unlikely given the processing differences. At least, we could not reasonably ascribe such a common bias to AS-off processing.

One way to explain the differences shown in Figures 4 and 5 is to note that the MERRA-2 comparisons occur for different sets

of profiles than the JPL-CDAAC comparisons. It is possible that different groups of profiles exhibit different biases, if the biases might be geographically or temporally dependent. We can rule out profile grouping as a factor by performing the comparisons to MERRA-2 with the same subset of profiles that we used in the JPL-CDAAC comparisons. Figure 6 plots the median differences with MERRA-2 using only the common set of profiles existing in the two pairs of data sets: (JPL–CDAAC AS-off) and (JPL–CDAAC AS-on). We label the profiles common among the JPL and CDAAC data sets as the "profile-to-

profile subset" or P2P subset, following (Ho et al., 2012). Figure 6 demonstrates that the differences between Figures 4 and 5 are not simply due to the different subsets of profiles used for the MERRA-2 comparisons than for the JPL-CDAAC comparisons. We also separated the JPL profiles by AS on and off within the P2P subset, and find very similar patterns to Figure 6 where AS on and AS off are combined for the JPL profiles. The two RO data sets are differenced at the coarser set of altitudes corresponding to the JPL data set (i.e. CDAAC profiles are interpolated to JPL altitudes from a finer set of vertical

levels). The CDAAC comparisons to MERRA-2 occur at a set of altitudes that generally do not match the JPL altitudes. Therefore, the median of the numerical differences formed for Figure 4 will not be exactly the same as the median of the numerical differences formed for Figure 6, although both figures involve common sets of profiles. An additional interpolation is performed for plotting the differences on a 1-km altitude grid.

In Figure 5, we compared all profiles that passed QC to MERRA-2. Between the altitudes of 10 km and 30 km, the biases between the retrievals and MERRA-2 show very minor differences whether all QC-passed profiles are used, or if the P2P subset is used. This suggests that the biases differences between Figures 4 and 5 are not due to the different locations of the retrievals used in these two comparisons.

**4 Discussion**

In this section, we discuss comparisons between JPL single-frequency processing, CDAAC AS-on and CDAAC AS-off dual-frequency processing, and the MERRA-2 reanalysis. Given the excellent agreement between the JPL and CDAAC data sets,

and larger disagreement between these data sets and MERRA-2, we suggest that the biases exceeding ~0.25% in Figure 5 are due to biases in the reanalysis, which appear to be altitude-dependent, reaching local maxima near 15 km and 17 km. This conclusion is reinforced by the fact these biases are similar for different sets of profiles used in the comparisons (compare Figures 5 and 6).

That MERRA-2 biases may be more significant near 15-17 km altitudes. near the tropical tropopause, does not contradict previous studies using MERRA-2. The MERRA-2 reanalysis, despite extending from the early 1980s (Basilovich et al., 2015) does not assimilate GPS radio occultation observations until 2004 (McCarty et al., 2016). Reanalysis temperatures near the tropopause are very dependent on the vertical resolution of the model at those altitudes (Tegtmeier et al., 2020). Because absolute accuracy of the reanalysis at these altitudes, particular before RO is assimilated, is difficult to characterize, some studies have used comparisons between different reanalyses to gain insight. Basilovich et al. (2015) shows that comparisons between MERRA-2 and ERA-Interim have the largest temperature differences in the tropical tropopause region at pressure levels between about 110-130 hPa, which is near altitudes of 15-17 km, and temperature differences approaching 1K (Figure 3-5 of that work). At these altitudes, a temperature difference of 1 K corresponds to a refractivity difference in magnitude of approximately 0.5%, which is close to the largest value observed here (Figure 5). A 15-year study comparing multi-mission GPS RO data to the ERA-Interim reanalysis spanning 2001-2016 finds that the refractivity differences are bounded by 0.2% in the altitude range 8-30 km (Gleisner et al., 2020). Larsen et al. (2005) report biases of up to 0.5 K maximizing near the tropopause for February 2000, compared to the ECMWF analyses from that period.

Biases between RO and MERRA-2 are also significant at altitudes above 20 km, although this only occurs for the JPL and CDAAC AS-on data sets, not the CDAAC AS-off data set. Variations in bias between the data sets are also observed at 15 km and 17 km. A possible reason for the bias differences between data sets is that possible biases in MERRA-2 are spatially and temporally dependent. The different locations of the profiles in the three data sets – where the comparisons are performed – could result in different biases. We view the data set differences in Figure 5, particularly the larger ones, as an opportunity to characterize the spatial/temporal bias variations that may exist in MERRA-2. Figure 5 suggests that such bias variations are altitude-dependent.

We have selected four altitudes to analyze possible latitude-dependent biases in the datasets.  The altitudes are 12 km, 15 km, 17 km and 26 km based on Figure 5. Figure 7 shows median biases between the various datasets partitioned across 15° latitude bins spanning 60S to 60N. Bias magnitudes < 0.25% are semi-transparent to emphasize bias magnitudes > 0.25%. The first three panels, which compare to MERRA-2, show the most significant bias variations with altitude occur at altitudes 15 km and 26 km. Biases at 15 km altitude broadly peak at low latitudes, possibly corresponding to the variation of tropopause height with latitude, with higher tropopauses at the lowest latitudes. The latitudinal behaviour of the biases at 15 km are significantly different than neighbouring heights 12 km and 17 km, possibly suggesting that the reanalysis in the period 1995-1997 is

constrained by observations differently near the tropopause than away from it. At the higher stratospheric altitude of 26 km, where RO becomes increasingly subject to residual biases due to ionospheric effects, all three data sets show similar latitudinal trends of more positive biases near the poles.


The bottommost panels of Figure 7 are comparisons between matched profiles between the JPL dataset and CDAAC AS-ON and OFF datasets. In both comparisons, the latitudinal dependence of the bias is much less than the MERRA-2 comparisons, regardless of altitude. This applies even to highest altitude of 26 km, where residual ionospheric effects should be largest. The full set of panels in Figure 7 are consistent with the interpretation that the latitude dependence is due to biases in the reanalysis

at altitudes 17 km and 26 km. The lack of RO bias at 26 km is perhaps not surprising, despite the different processing, because of the solar minimum conditions during 1995-1997 when these data were acquired.

## 4.1 Wider application of single-frequency processing

Single-frequency processing was developed for this paper to provide insight on the robustness of the complete GPS/MET

datasets from 1995-1997, as published at the CDAAC site. These data are now augmented by a single-frequency processed data set made available with this publication. In this section, we discuss other applications where single frequency processing may be useful.

In the early days of RO, dual-frequency processing was more challenging because so-called codeless processing was not yet

implemented that increases L2 SNR even in the presence of Anti-Spoofing encryption (Kursinski et al., 1996). Single frequency processing has been applied to the Oersted RO dataset for this reason (Larsen et al., 2005). Even now, the second GNSS frequency is only tracked at altitudes above 10-20 km because the L2 frequency is weaker due to encryption and often too weak to track at lower altitudes. Instead, extrapolation of the dual-frequency correction computed at higher altitudes is performed below this cutoff, which can incur ionospheric-dependent biases when the extrapolation is not accurate due to

ionospheric structure. At these lower altitudes, the effect of residual ionosphere is decreasing (Mannucci et al., 2006) but there may be instances near solar maximum where further analysis of this extrapolation error is warranted.

The GRAS-2 receivers planned for MetOp-SG (Second Generation) do not track the legacy encrypted L2 signals, since there are sufficient numbers of satellites transmitting the newer unencrypted dual-frequency signals. However, single frequency

techniques could be used on the remaining satellites from the GPS system that do not yet transmit the newer signals, of which there may still be some for several more years.

Another application where single-frequency techniques may be useful is investigation of residual systematic effects remaining after the standard dual-frequency correction is applied. The residual ionospheric error ($RIE_{DF}$) is primarily caused by the

differences between the L1 and L2 signal path trajectories relative to their trajectories in the absence of the ionosphere (Syndergaard, 2000). The L1 and L2 signal paths sample different parts of the ionosphere so that dual-frequency processing does not remove all ionospheric influence (Vorob'ev and Krasil'nikova, 1994; Syndergaard, 2000). Single frequency processing also is subject to a residual ionospheric error ($RIE_{SF}$) but its magnitude may be a factor of ~1.6 smaller than $RIE_{DF}$ if advanced ionospheric correction techniques are applied. This is discussed in the Appendix.


Comparing the single and dual-frequency correction methods may provide some information on the magnitude of the residual due to L1/L2 raypath separation to augment simulation studies of this error source (Danzer et al., 2013; Danzer et al. 2020). Danzer et al. (2013) estimate that a bending angle bias due to raypath separation in the ionosphere can reach ~0.4 $\mu$rad near 30 km altitude for solar maximum daytime conditions, which corresponds to a fractional bending angle error of approximately

0.1%.. The analysis in this paper suggests that single-frequency retrieval error bias is less than 0.25% in *refractivity* for solar activity conditions closer to minimum (1995-1997). As noted by Danzer et al. (2013), residual ionospheric errors decrease in fraction by about a factor of three going from refractivity to bending angle, so that the bending angle bias corresponding to the refractivity bias found here may be close to the estimate of 0.1% in Danzer et al. (2013).

We performed an estimate of the bending angle precision of single-frequency processing by differencing bending angles retrieved by single and dual-frequency processing during AS-off conditions, using the JPL processing chain. The results are shown in Figure 8 for approximately 2000 profiles. The estimates of standard deviation of the bending angle differences between single- and dual-frequency processing are shown versus impact altitude in the green curve. The 1$\sigma$ bending angle precision of ~10-15 $\mu$rad near 30 km altitude, arising due to random error and applicable to single profiles, is

approximately a factor of 2-3 larger than has been estimated for CHAMP and GRACE radio occultation measurements (Healy et al., 2007), which are expected to have higher precision than GPS/MET. Resolving differences between the dual- and single-frequency residual ionospheric errors using single frequency processing may require averaging ~1000 daytime profiles to reduce the random component of error affecting single profiles, assuming that the random error decreases as $\sqrt{N}$ where $N$ is the number of profiles in an ensemble average. Such an ensemble would reduce

random error to acceptable levels so that a bias of ~0.4 $\mu$rad could be resolved. Such numbers of profiles are readily available from the COSMIC-2 constellation as well as commercial providers of radio occultation data.

Healy and Culverwell (2015) proposed the so-called "kappa-correction" algorithm to reduce the magnitude of $RIE_{DF}$ by estimating the residual based on a formula in the Vorob'ev and Krasil'nikova (1994) paper. Evaluation of this correction term

has been promising (Danzer et al., 2015; Danzer et al., 2020). Efficacy of the kappa correction varies with satellite altitude. For satellites in lower orbits (e.g. the COSMIC-2 satellites at ~530 km versus ~800 km for COSMIC-1) the reduction of $RIE_{DF}$ is less effective. As shown in the study by Mannucci et al. (2011), there is an error cancellation effect for satellite altitudes

well above the height of peak ionospheric vertical gradients, which typically occur just below the altitude of peak electron density (~300 km–400 km altitude). The occulting ray first enters the ionosphere from above after leaving the GNSS transmitter. Then the signal exits the ionosphere and re-enters from below as it propagates to the LEO. If the LEO is orbiting in the ionospheric topside (well above the F-layer density peak), then the residual error approximately cancels between the entrance and exit phases, although not fully because of ionospheric differences between entry and exit regions. Additional insights are available in Liu et al., (2020), Danzer et al. (2020) and Li et al. (2020).

## 5 Conclusions

We have performed an analysis of reprocessed GPS/MET data spanning 1995-1997 generated by CDAAC in 2007. CDAAC developed modified dual-frequency processing methods for the encrypted data (AS-on) during 1995-1997. We compared the CDAAC data set to the MERRA-2 reanalysis, separately for AS-on and AS-off, focusing on the altitude range 10-30 km. MERRA-2 did not assimilate GPS/MET data in the period 1995-1997. To gain insight into the CDAAC data set, we developed a single-frequency data set for GPS/MET, which is unaffected by the presence of encryption. We find excellent agreement between the more limited single frequency data set and the CDAAC data set: the bias between these two data sets is consistently less than 0.25% in refractivity, whether AS is on or not. Given the different techniques applied between the CDAAC and JPL data sets, agreement suggests that the CDAAC AS-on processing and the single frequency processing are not biased in an aggregate sense greater than 0.25% in refractivity, which corresponds approximately to a temperature bias less than 0.5 K.

Since the profiles contained in the new single frequency data set are not a subset of the CDAAC profiles, the combination of the CDAAC data set, consisting of 9,579 profiles, and the new single-frequency data set, consisting of 4,729 profiles, yields a total number of 11,531 unique profiles from combining the JPL and CDAAC data sets. All numbers refer to quality control having been applied.

We performed comparisons between the GPS/MET data sets and the MERRA-2 reanalysis. The biases between the observations and MERRA-2 exceeded the biases between the JPL and CDAAC data sets, suggesting possible altitude-dependent biases in MERRA-2. The CDAAC AS-off data set generally had the smallest bias with respect to MERRA-2, although increased biases appeared at 15 km altitude and near 30 km altitude. The bias between MERRA-2 and the CDAAC AS-on data set reached a local peak near 17 km altitude, starting to increase above 18 km altitude, eventually reaching nearly 0.75% difference. Similarly, the JPL data set bias increased above 18 km altitude, and exhibited a local peak near 15 km.

Considering that the differences between MERRA-2 and the three data sets (JPL, CDAAC AS-off and CDAAC AS-on) are not identical, it is possible that the different times and locations of the three data sets could be responsible for the varying differences with MERRA-2. Differences between observations and MERRA-2 were also computed for a limited set of

occultation events that were common between JPL and CDAAC AS-on, and between JPL and CDAAC AS-off. The biases between observations and MERRA-2 were similar whether a common set of profiles were used or not. This suggests that the locations of the radio occultation profiles were not a significant determining factor in the bias differences between the retrievals and MERRA-2.

We examined the biases between the retrievals and MERRA-2 as a function of latitude band and altitude. At 15 km altitude, the biases between all three retrieval sets and MERRA-2 have a broad peak near low latitudes. The opposite trend is seen at 26 km altitude: the biases are largest at the high latitudes (north and south). At 17 km, only the JPL data set shows significant variation with latitude. In contrast, there is less variation of bias with latitude between JPL and the two CDAAC data sets (AS-on, AS-off).


     Single frequency processing of RO data may be a useful method to compare with dual-frequency methods, pending further analysis. Both single- and dual-frequency processing methods are subject to a residual ionospheric error or RIE. However, the residual error for single-frequency processing is potentially a factor of ~1.6 smaller than the residual for dual-frequency processing if estimates of ionospheric electron density structure are used in the retrieval, as discussed in Syndergaard (2000).

Although other errors can affect single frequency processing, such as code multipath, these errors are likely independent of ionospheric conditions. Therefore, ensembles of profiles using single frequency processing could be used as a means of validating corrections applied to reduce dual-frequency RIE. Single-frequency processing should first be validated against dual-frequency profiles during periods when the RIE is negligible, e.g. at nighttime or during solar minimum. Single frequency processing holds promise when two frequencies are unavailable or as a means of validating L2 processing when a new receiver

is being evaluated that uses special L2 processing to overcome encryption. Two previous studies discuss the potential value of single frequency processing (Larsen et al., 2005; de la Torre Juarez et al., 2004).

     Based on these results, future reanalyses should consider using the full combined data set of single- and dual-frequency processed data from GPS/MET, consisting of 11,531 unique profiles. Healy et al. (2017) found positive impact on the ERA5 reanalysis after assimilating UCAR GPS/MET data from a limited AS-on period spanning December 23, 1996 to January 13,


1997.

## 6 Data Availability

     The UCAR CDAAC data set can be found at: http://cdaac-www.cosmic.ucar.edu. The new single-frequency data set can be

found at https://doi.org/10.48588/jpl.221447.

## 7 Author Contribution

A. J. Mannucci led the conceptualization, investigation and methodology, as well as contributing to validation and analysis software. C. Ao contributed to the methodology, validation and software. B. Iijima contributed to the investigation and software used in this work. T. Meehan contributed ideas to the investigation. P. Vergados contributed to the investigation and provided useful review and editing of the manuscript. E. R. Kursinski contributed to the methodology, validation, and provided review and editing. W. S. Schreiner provided inputs on the methodology and validation, and review and editing.

## 8 Competing Interests

The authors declare that they have no conflict of interest.

## 9 Acknowledgements

Portions of this research were carried out at the Jet Propulsion Laboratory, California Institute of Technology, under a contract with the National Aeronautics and Space Administration. Support of the NASA Earth Science Division is acknowledged. The lead author would like to acknowledge discussions with S. Healy of ECMWF that influenced the direction of this work.

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

**Appendix: Comparison of Residual Ionospheric Errors Between Single- and Dual-Frequency Retrievals**

In this appendix we compare residual ionospheric error between dual- and single-frequency processing. We adopt the notational conventions of Syndergaard (2000; hereafter referred to as S2000) and provide a summary of their approach. The key aspects of the approach are to: 1) account for the fact that the ionosphere alters the signal path trajectories and 2) define residual ionospheric error relative to a hypothetical situation where all signal bending and delay is caused by the neutral atmosphere.

The total refractive index of the atmosphere and ionosphere is given by: $1 + N_n + N_i(f)$ where $N_n$ is the refractivity of the neutral atmosphere and $N_i(f)$ is the refractivity of the ionosphere, which depends on the frequency $f$ as follows: $N_i(f) = -C\frac{N_e}{f^2}$, where $N_e$ is the ionospheric electron density (number per m³) and $C$ is 40.3 m³/s². The measured phase path for the L1 signal is given by:

$$L_1 = \int_1 [1 + N_n + N_i(f_1)]ds \qquad \text{(A1)}$$

where the integral is along the trajectory of the L1 signal. The ionospheric correction derived from phase and range data (Equation (3)) is proportional to $\gamma_1$, the integral of electron density along the path taken by L1:

$$\gamma_1 = \int_1 N_e ds \qquad \text{(A2)}$$

The ionosphere-corrected phase path $L_C$ used in single frequency retrievals is:

$$L_C = L_1 - \frac{-C}{f_1^2}\gamma_1 \qquad \text{(A3)}$$

The true "ionosphere-free" phase observable $L_F$ is defined as:

$$L_F = \int_F [1 + N_n]ds \qquad \text{(A4)}$$

which is the integral of neutral atmosphere refractive index along a fictitious signal path where the ionosphere is absent. The
730 residual ionospheric error RIE$_\text{SF}$ for single-frequency processing is thus:

$$\text{RIE}_\text{SF} = L_C - L_F = \int_1 [1 + N_n]ds - \int_F [1 + N_n]ds \qquad \text{(A5)}$$

RIE$_\text{SF}$ is caused by the difference between the actual path traveled by L1 and the ideal path that would occur in the absence of the ionosphere. RIE$_\text{SF}$ is identical to Equation (17) in S2000 (= $\rho_1 - \rho_F$ using their notation).

It turns out that the expression for the dual-frequency residual ionospheric error (Equation 18 of S2000) has identical terms to
735 Equation (17) of S2000. In Equation (17) of S2000, each term is either a factor of 1.6 smaller, or a factor of 2.2 smaller, than the corresponding term in Equation (18). Note, however, that Equation (18) of S2000 is the RIE using the phase-path

ionospheric correction approach, which is known to be less accurate than the standard bending angle correction approach. S2000 discusses means of estimating ionospheric electron density information to reduce the RIE when the phase path correction method is used. These approaches would also reduce $RIE_{SF}$, since SF processing is also a phase path correction method. Such approaches result in an RIE that is comparable to the bending-angle-based ionospheric correction. Therefore, the ionospheric estimation methods suggested in S2000 applied to single-frequency processing could produce an $RIE_{SF}$ that is a factor of 1.6 smaller than the standard bending angle-based $RIE_{DF}$. Improvements to the standard dual-frequency correction, such as discussed by Healy and Culverwell (2015), would likely produce residuals smaller than what is straightforward to achieve with SF processing.

For completeness, we now examine errors of single-frequency processing associated with the approximations made in Equations (1)-(3) that assume the ionospheric effect on the phase and pseudorange paths depends on the inverse square of the frequency. These approximations ignore what are known as "higher-order terms" caused by the ionosphere. We compare higher-order terms associated with single- and dual-frequency processing. Using the notation in Bassiri and Hajj (1993; henceforth referred to as BH93), the single-frequency combination of Equation (3) would lead to the following higher-order terms, up to third-order (using BH93's Equations 8.a and 9.a):

$$0.5 \, (P_1 - L_1) = 0.5 \left( \frac{q}{f_1^2} + \frac{3s}{2f_1^3} + \frac{4r}{3f_1^4} \right) \qquad (A1)$$

where $q/2$ is denoted as $I$ in Equation (3) and $s$ and $r$ are defined in BH93 as integrals over the ionospheric electron density, density squared, and magnetic field along the radio occultation raypath (Equations (11) and (12) of that paper). Our purpose here is not to estimate these terms, but to compare the single-frequency and dual-frequency higher-order terms, which share these same integral formulations for $s$ and $r$.

For the dual-frequency combination, the higher-order terms have the following multipliers for $s$ and $r$ (BH93 Equation 17.b):

$$s : \frac{1}{2f_1 f_2 (f_2 + f_1)}, r : \frac{1}{3f_1^2 f_2^2} \qquad (A2)$$

Table A1 shows the magnitudes for the multipliers of these higher order terms and their ratio. Note that the factors $s$ and $r$ are frequency-independent.

**Table A1.** Magnitudes of different higher-order ionospheric terms. Units in powers of Hz.

| Multiplier Term | Magnitude | Ratio single to dual frequency |
|---|---|---|
| $s$, single-frequency | $1.92 \times 10^{-28}$ | 2.1 |
| $s$, dual-frequency | $9.22 \times 10^{-29}$ | |
| $r$, single-frequency | $1.08 \times 10^{-37}$ | 1.2 |
| $r$, dual-frequency | $8.91 \times 10^{-38}$ | |


Whereas the higher-order term for single-frequency processing is up to a factor of two larger than the corresponding term for dual-frequency processing, the signal path bending effects discussed earlier in this appendix are the dominant contributors to $RIE_{DF}$ and $RIE_{SF}$, exceeding by many factors the second-order term associated with factor $s$. S2000 identifies the raypath bending effect as the "dispersion residual" and does not otherwise consider higher-order terms corresponding to $s$ and $r$ in
their analysis of RIE.

Quantifying the dispersion residual requires simulating realistic ionospheric electron density distributions and computing integrals along realistic RO raypath trajectories that take raypath bending into account. These simulations can also be used to compute the factors $s$ and $r$ once a model for the geomagnetic field is adopted. An early study by Hardy et al. (1994) for
daytime conditions near solar maximum shows the dispersion residual to be up to a factor of 10 larger than the "second-order" residual associated with factor $s$ at locations along the raypath corresponding to altitudes near 100 km (their Figure 9; the contribution of the third-order residual $r$ is negligible). At lower altitudes, the magnitude of the dispersion residual gradually decreases and can be comparable to the second-order residual near 30 km altitude, but the integrated effect from higher altitudes will dominate the retrieval. Similar conclusions are found in the simulations by Melbourne et al. (1994, Figures 8-19 and 8-
20). Hoque and Jakowski (2010) simulate daytime conditions for two cases near solar maximum and find that the maximum excess phase path contributed by raypath bending far exceeds the second order residual (see their Table 1. The bending term is denoted as "excess path length"). The bending term is 98.5 and 271 cm for L1 and L2 frequencies, respectively. The second-order term contributes 6.2 cm of excess path at L1. It is interesting to note that the late afternoon ionosphere (their "Case 2"), while containing fewer electrons than the noon-time case, contributes more excess path due to bending because of the vertical
electron density gradients in the lower ionosphere.

The analysis in this Appendix and prior research confirm that single-frequency retrievals can play a role in analysing $RIE_{DF}$. However, care is required to carefully consider the ensemble of profiles that is used to do the analysis since $RIE_{DF}$ is strongly affected by ionospheric electron density vertical profiles. We note that this conclusion is valid for the radio occultation
geometry, but is less applicable to ground-based GNSS observations, where RIE tends to be less sensitive to vertical electron density gradients.

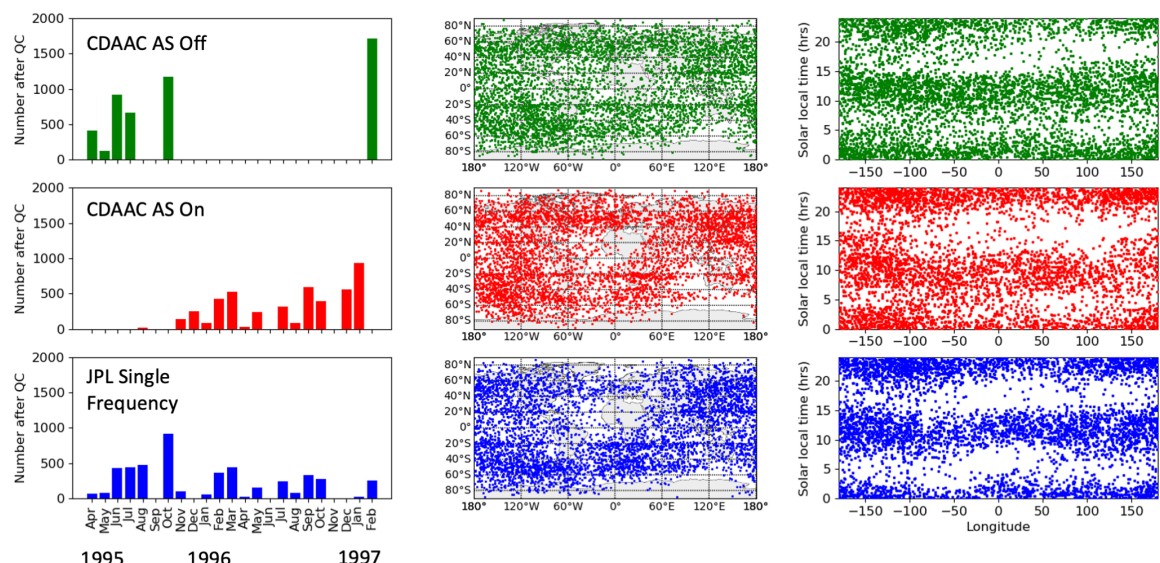


**Figure 1. Information on the three data sets analyzed: UCAR's CDAAC AS-off, AS-on and JPL single frequency. The left column shows the number of profiles by month, after quality control is applied. The middle column shows the locations of the profiles. The right column shows the solar local time of the profiles versus longitude.**


**Fitting step:**

1. Define overlapping time intervals (90%)
2. Fit polynomial over each interval

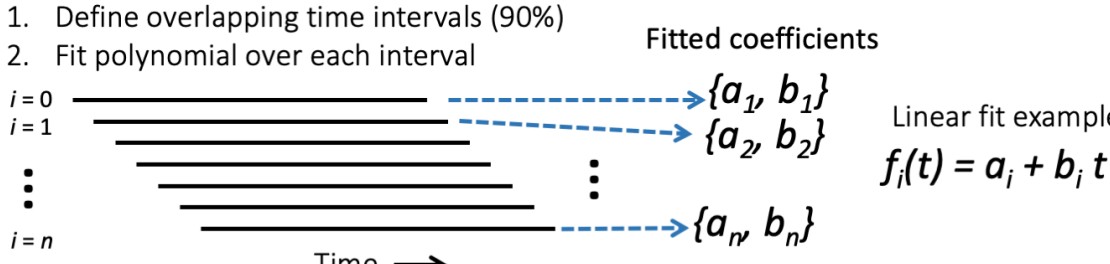

$i = 0$
$i = 1$
$\vdots$
$i = n$

Time $\longrightarrow$

**Fitted coefficients**

$\{a_1, b_1\}$
$\{a_2, b_2\}$
$\vdots$
$\{a_n, b_n\}$

Linear fit example

$f_i(t) = a_i + b_i\, t$

**Evaluation step:**

1. Evaluate each fit $i$ at time $t$
2. Perform a weighted sum over each evaluation

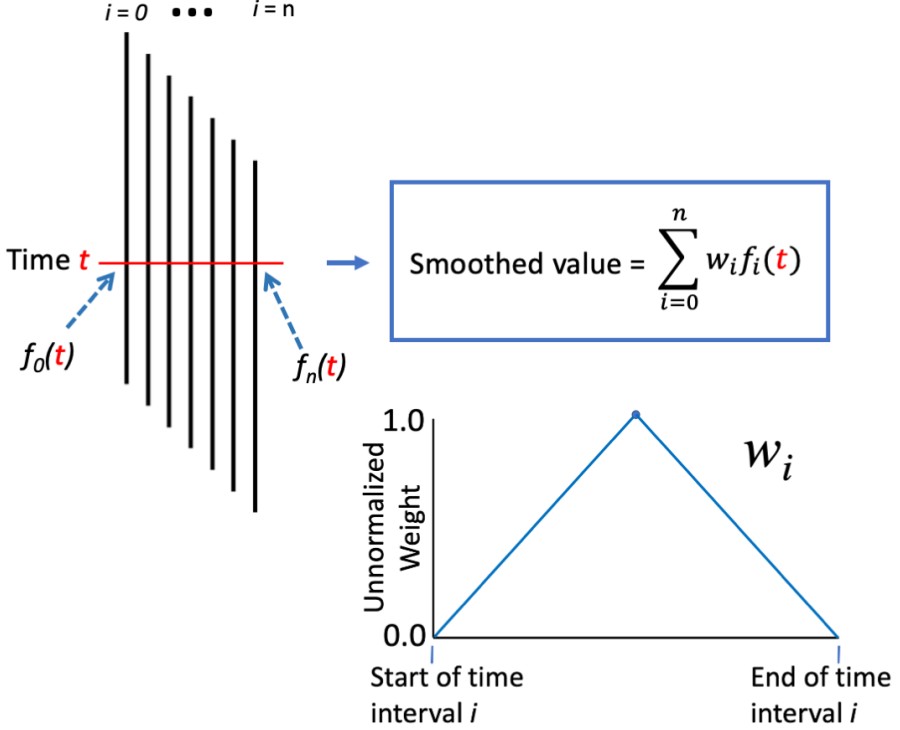

$i = 0$ $\bullet\bullet\bullet$ $i = n$

Time $t$

$f_0(t)$ $\qquad$ $f_n(t)$

Smoothed value = $\displaystyle\sum_{i=0}^{n} w_i f_i(t)$

$w_i$

1.0

0.0

Unnormalized Weight

Start of time interval $i$ $\qquad$ End of time interval $i$

**Figure 2. Illustration of the smoothing process used in single-frequency processing. This example shows linear fits to the segments, although quadratic fits were actually used.**

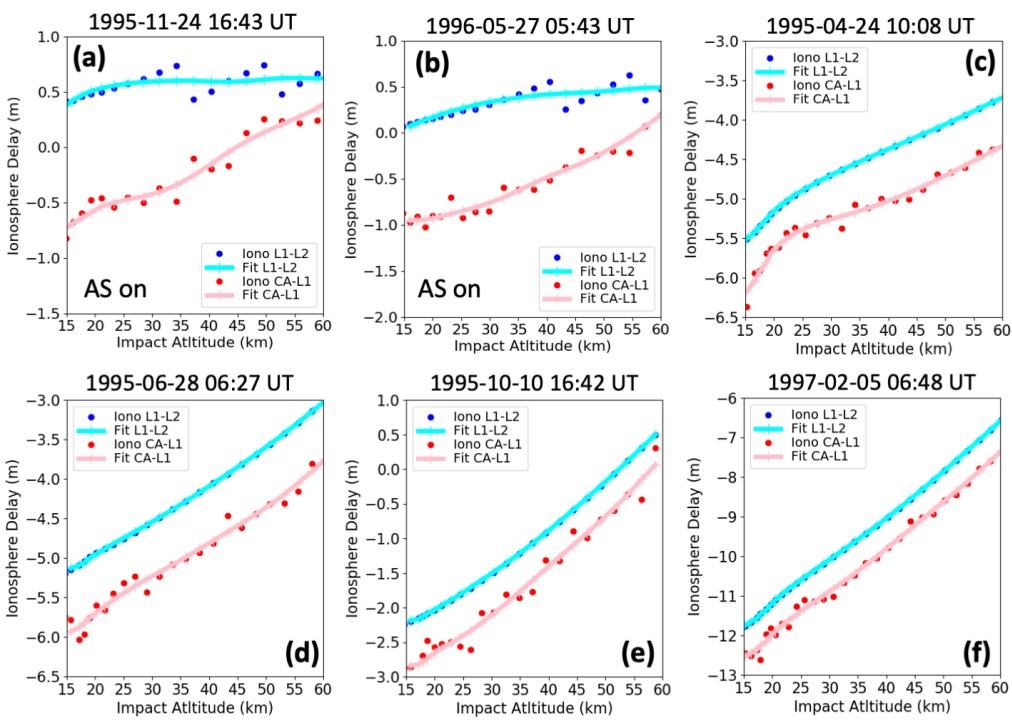


**Figure 3. Examples of ionospheric pseudorange path changes (converted to delay in meters) using the single-frequency method and the smoothing approach illustrated in Figure 2 (pink). Panels (a) and (b) are examples where AS is on, and the L2 frequency is much weaker. Panels (c)-(f) correspond to AS-off, and standard dual-frequency techniques could be used. Ionospheric contributions from dual-frequency processing are shown for comparison (blue).**


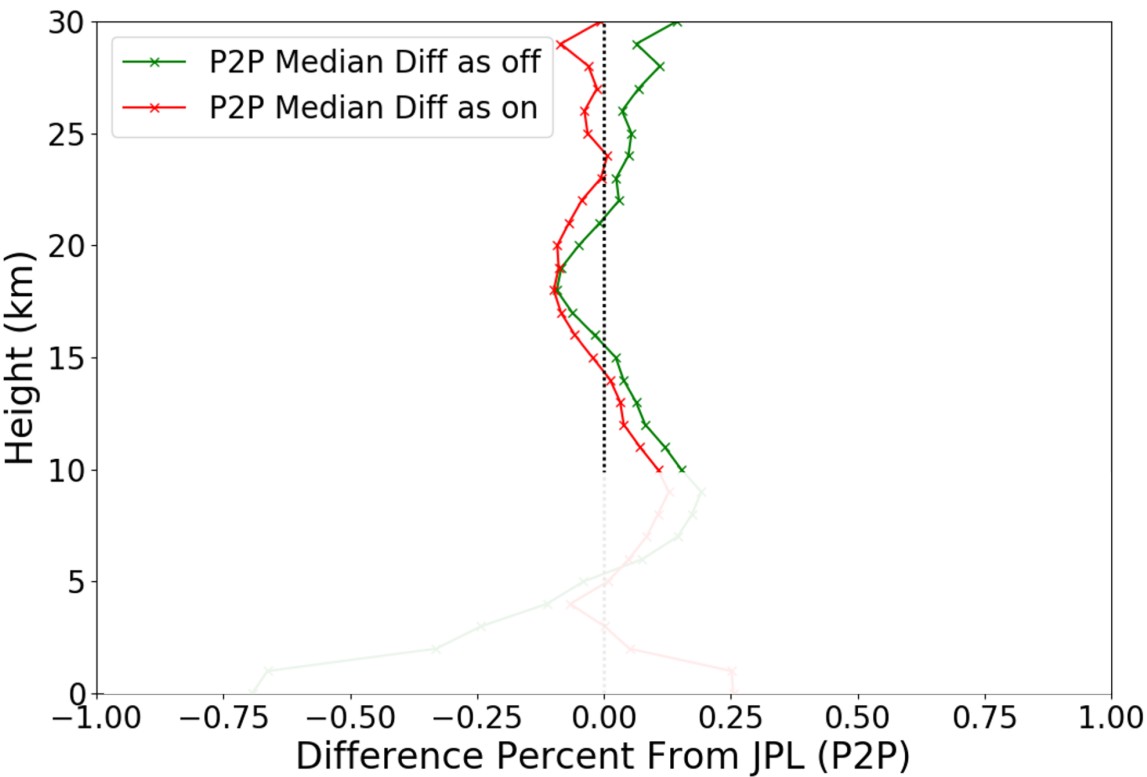

**Figure 4. Percentage refractivity median difference of JPL minus UCAR AS-on and AS-off for profiles common to the**

**single frequency and UCAR data sets ("P2P" data set). The focus altitudes are 10 km-30 km. Profiles are interpolated**

**to a common 1-km vertical grid spacing. Percent difference is with respect to the mean refractivity at the corresponding**

**altitude.**

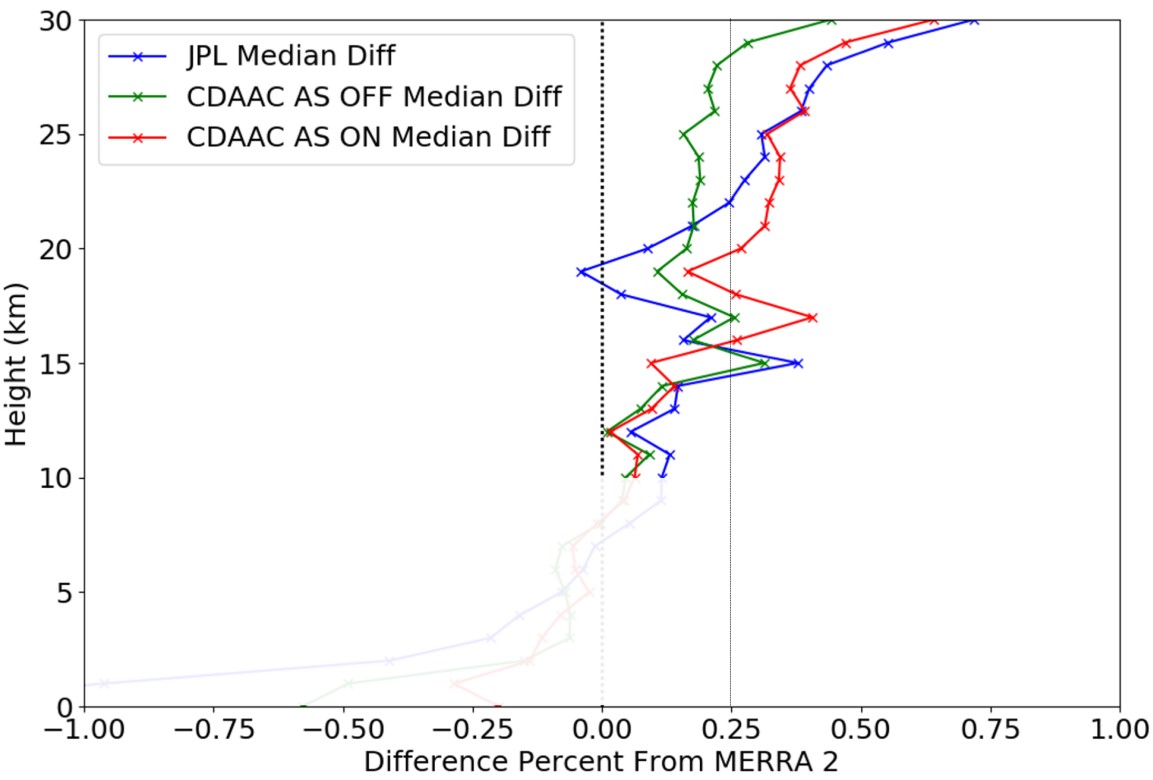


**Figure 5. Percentage refractivity median differences of the JPL and CDAAC data sets minus the MERRA-2 reanalysis, interpolated to a common 1-km altitude grid. Altitudes between 10 and 30 km are analyzed. Percent difference is with respect to the mean MERRA-2 refractivity at the corresponding altitude.**


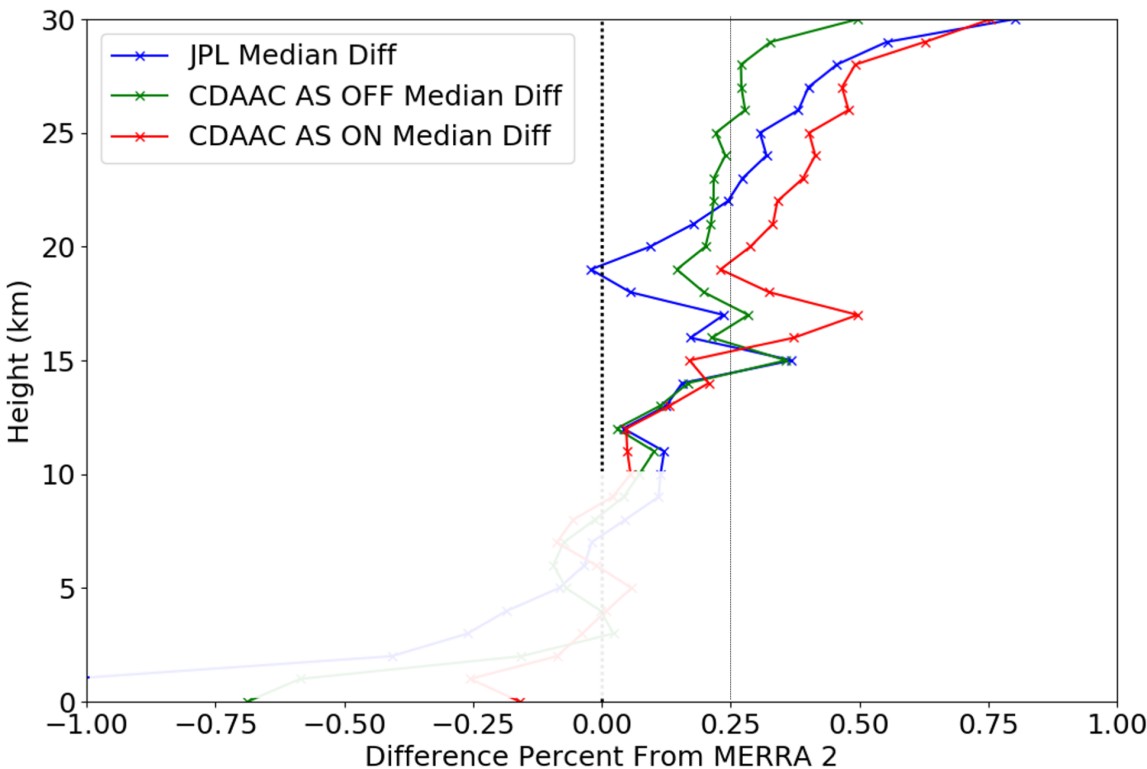

**Figure 6. Similar to Figure 5 except using only the set of profiles common between the JPL and CDAAC data sets that is the same set used in the comparison shown in Figure 4. Percent difference is with respect to the mean MERRA-2 refractivity at the corresponding altitude.**


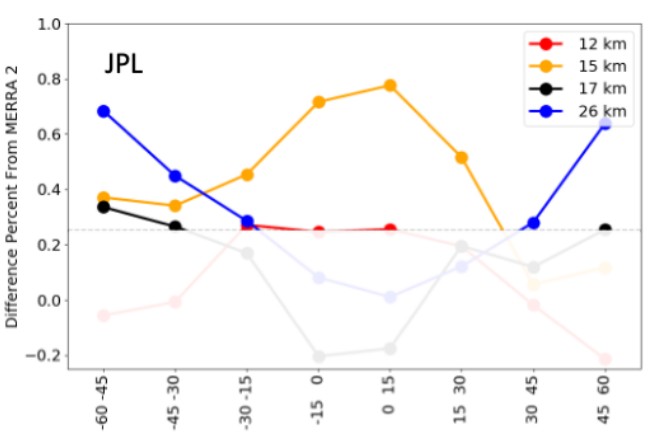

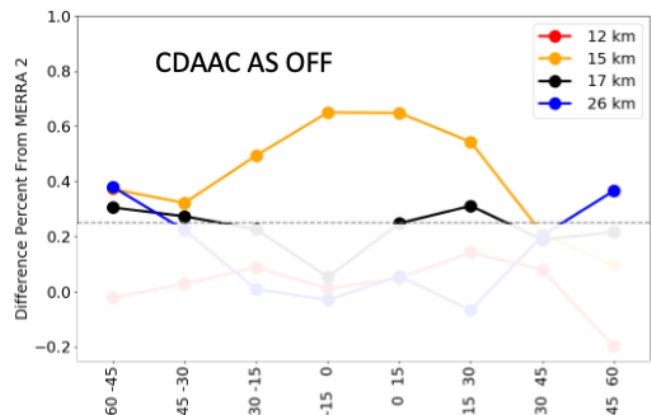

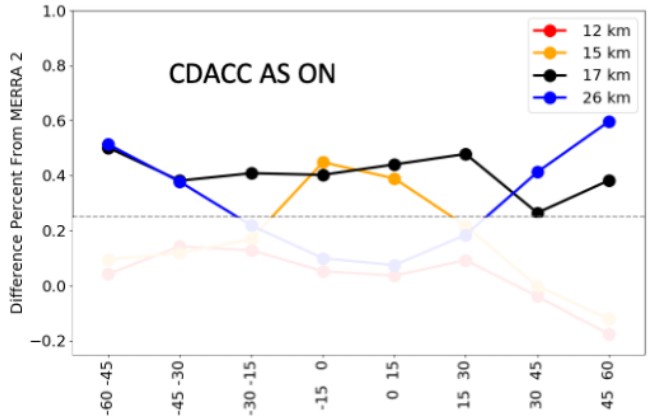

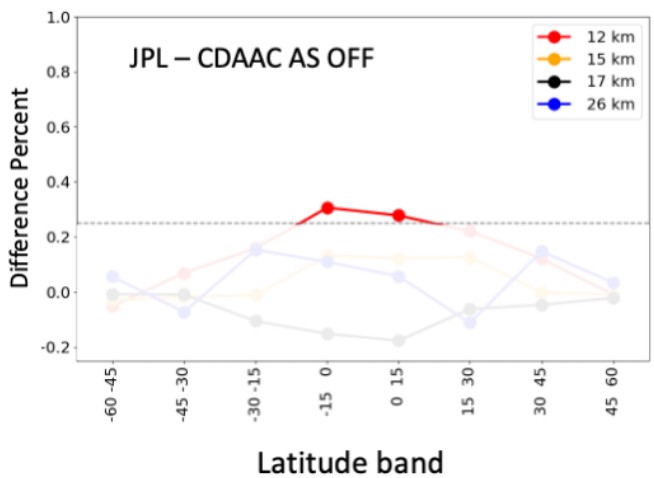

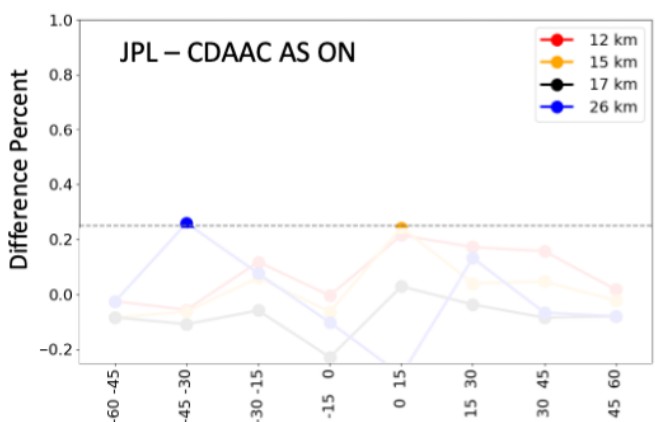

**Figure 7. Latitude dependent biases between different data sets for a series of representative altitudes (12, 15, 17 and 26 km). Latitude band in degrees is along the x-axis for all panels. The top row shows differences of the JPL and CDAAC AS-off data sets minus MERRA-2, for different latitude bands and different altitudes. The middle row shows the differences of CDAAC AS-on minus MERRA-2. The bottom row shows differences of JPL single frequency minus CDAAC AS-off and AS-on data sets, for different latitude bands and altitudes. The figure areas where difference magnitudes are < 0.25% are semi-transparent to emphasize difference magnitudes > 0.25%.**



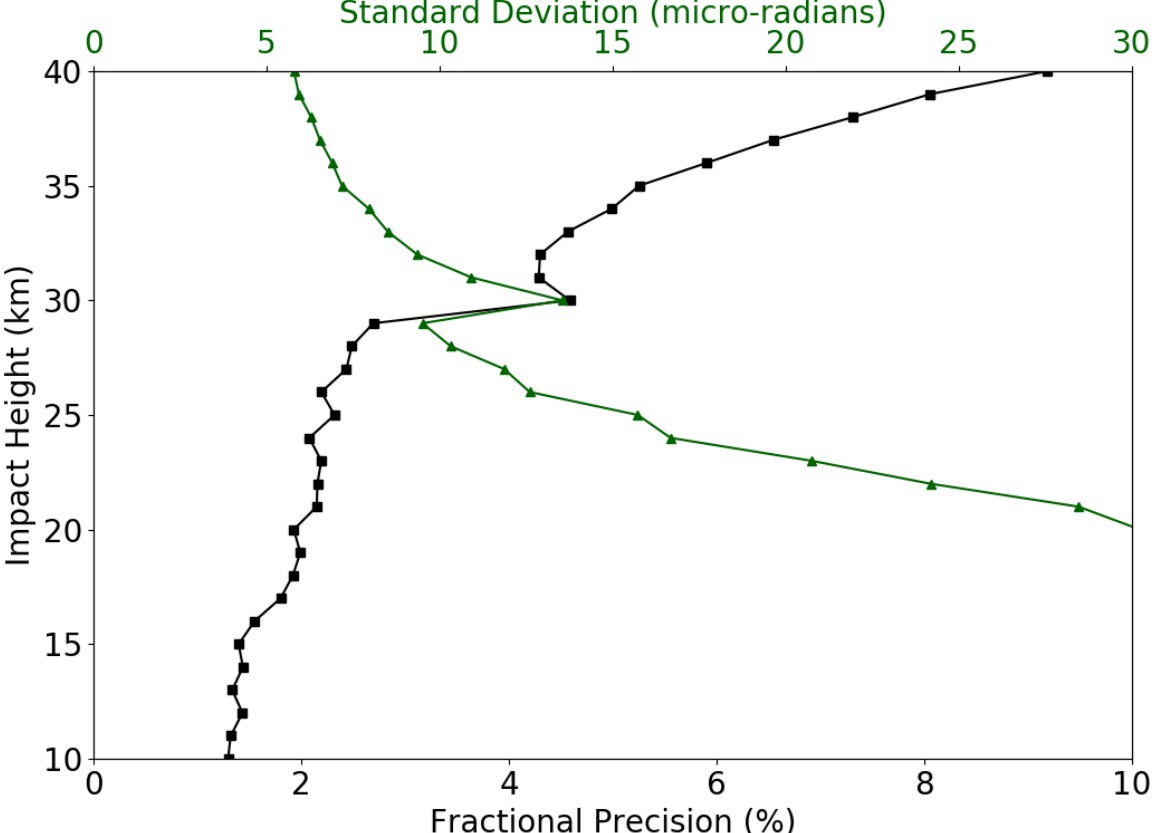

**Figure 8. Estimates of the standard deviation of the differences (green curve) between single- and dual-frequency bending angle for AS-off conditions using the JPL radio occultation processing chain, for approximately 2000 profiles. The black curve is the precision divided by the mean refractivity at the corresponding altitude. Precision is estimated using the robust statistic of interquartile range divided by 1.349, which corresponds to $1\sigma$.**
