# Peer review of "An Assessment of Reprocessed GPS/MET Observations Spanning 1995-1997"

_Atmospheric Measurement Techniques, 2021_

## Author Comment (AC1)

**General comment:**

The paper addresses an analysis of reprocessed GPS/MET data using a single frequency retrieval technique which is compared to the dual frequency processing product of CDAAC 2007 and the MERRA-2 analysis. With this new and partly not overlapping dataset they can extend the CDAAC 2007 dataset and hence can provide more (RO) measurement data for climate trend analysis and climate model validation in a time where globally even distributed measurement data is sparse.

The paper is structured in a good way, the title clearly reflects the contents of the paper, and the scientific methods and assumptions are valid and clearly outlined. The results are sufficient to support the interpretations and conclusions. Some minor questions, comments and typos I will provide in the specific comments.

We appreciate the reviewer's careful reading of our paper. We address every specific comment below.

**Specific comments:**

1. Section 2 page 4 line 121: I would change "... and underwent reprocessing in 2007." to "and underwent **a first** reprocessing in 2007." since there were more than this reprocessing in the last years.

We have addressed this comment and also clarified that the previous sentence referred to GPS/MET data.

2. Section 2.1, page 5 line 143/144: I would add the cite to Liu et al. 2020 https://doi.org/10.3390/rs12213637 ("New Higher-Order Correction of GNSS RO Bending Angles Accounting for Ionospheric Asymmetry: Evaluation of Performance and Added Value") on the bi-local higher-order RIE here and also in Section 4.1 page 15 line 466.

We appreciate the reviewer bringing this reference to our attention. We agree it is an appropriate reference and has been inserted.

3. Section 2.1 page 6 last paragraph: Which orbit/orbit processing software did you use? are there any updates to the CDAAC 2007 processing? Could you add a short sentence to specify this in your paper?

In that paragraph, we have added more specific details and references to the JPL and CDAAC processing.

4. Section 2.1 page 7 line 216ff: Could you please explain here a little bit more detailed what you mean by this sub-interval processing. What I think I understand from the

following paragraphs and Figure 2 you're doing this on each "ray" for the 1 Hz pseudo-range data and then interpolating this to the 50 Hz CA measurements. But I'm not sure that I've understand that correctly.

Thank you for the comment. We agree that the description was not sufficiently clear. We have attempted to clarify. Please let us know if questions remain.

5.  Section 2.1 page 8 line 233: How do you extrapolate below 15 km? constant, linear, ...?

We have clarified that the extrapolation is linear.

6.  Section 2.1 page 8 line 254/255: The sentence "The altitude range for the fits in 15-60 km, or whatever the upper altitude of the occultation happens to be." is finishing without being finished ...

Thank you for pointing out this awkward sentence construction. We have made changes.

7.  Section 2.1 page 8/9 line 258 - 260: I think these two sentences can be removed since in the next paragraph you're telling the same only in more detail.

Done.

8.  Section 2.1 page 9 line 275: typo: "...AS **in** on." -> "... AS **is** on."

Done.

9.  Section 3 page 11 line 331f: "Other climate-related work that ...": please add a cite which work you mean.

We have added references and clarified our meaning.

10. Section 3 page 12 360ff and Figure 6: To compare the CDAAC AS on AS off data with the new JPL data it would be good to separate the JPL data according to AS on and AS off too, although there is no difference in the processing in the JPL case.

Thank you for the suggestion. We did separate the JPL data into AS on and AS off and show the figure here. However, we don't believe that it adds significantly new material to the presentation, so we simply add some text in the paper to this affect.

[Figure]

Variation of Figure 6 with AS on (■) and AS off (▲) plotted for JPL data, along with AS ON and OFF combined (x).

11. Section 3 page 12 line 371: typo: remove the "**.**" after 30 km.

We find a comma "," there rather than period. Please let us know if that is not appropriate.

12. Section 4 page 13 discussion on Figure 7 lines 406ff: The bias of the 12 km level is larger at higher (shown) latitudes (JPL dataset but also both CDAAC datasets). This could be a problem of the MERRA-2 analysis there. The approximate height of the mid-latitude tropopause should be there. Could these biases at this height level be related to the mid-latitude tropopause? You only mention the tropopause with respect to the lower latitudes and the tropical tropopause. Please discuss this. The 17 km level shows an opposite effect than the 15 km level for the JPL data. This could indicate that there is a possible mislocation of the tropopause.

Thank you for your thoughtful comments. The reason that we only discuss the lower latitude tropopause is because we are focused on differences greater than 0.25%, because we cannot claim that difference magnitudes <= 0.25 % are significant (based on Figure 4). In reviewing this section and the figure, we noticed there is room for improvement. Our discussion focuses on biases that exceed 0.25%, as biases less than this magnitude may be due to measurement error. Therefore, we have redrawn the figures to emphasize the region where percent differences are

within 0.25%. We have changed the figure caption accordingly and modified corresponding text in the paper.

13. Section 5 page 16 line 503 typo/auto-correction error: "...radio **location** ..." -> "radio **occultation**"

Fixed.

14. Figure 4, Figure 5, and Figure 6: no x unit.

We believe the x-unit is properly labeled as "Percent difference from …". We have added some detail in the caption to be clearer.

15. Figure 7: caption: you could interchange AS-on and AS-off in the caption since then it corresponds to the panel order.

Thank you. Done.

---

## Author Comment (AC2)

General comments:

The paper compares radio occultation GPS/MET data retrievals from UCAR/CDAAC dual-frequency processing in 2007 with a newly processed GPS/MET dataset from JPL using single-frequency processing. The datasets partly overlap, and both contain periods with AS-on and AS-off. Together the datasets provide more data than before in the GPS/MET period of 1995-1997.  Since the JPL dataset relies on L1 single-frequency processing, it is insensitive to the L2 encryption (AS-on).  On the other hand, the single-frequency processing is affected by psudorange noise.  All datasets are compared to MERRA-2 reanalyses in the 10-30 km range, concluding that errors at some altitudes and latitudes in MERRA-2 are likely larger than the errors from either approaches of retrieving the refractivity from the GPS/MET data.  Thus, it is likely that future reanalysis projects could benefit from single-frequency processed GPS/MET data.

At times I found it difficult to understand exactly how the single-frequency processing is done, and I think a few additional equations may help (see specific comments below). Also, parts of the explanations seem to be contradictory, but I think it is just a question of being more precise in the language.

At the end, a possibly use of single-frequency processing is discussed, suggesting that it may provide some information on the magnitude of the residual ionospheric error due to raypath separation in dual-frequency processing. I have some doubt on that, partly because I don't think the single-frequency processing avoids raypath-induced residual ionospheric errors (see specific comment below), and partly because I think errors from pseudorange noise may be too large to be able to conclude much about residual ionospheric errors. However, I think it would be relatively easy to estimate the errors in bending angle due to the pseudorange noise, and I suggest a simple way to do that at the end of this review. This could then be compared to the expected size of residual ionospheric errors.

We thank the reviewer for a careful reading of the paper and the corresponding comments, which we address below, in an attempt to improve the manuscript.

Specific comments:

There are many acronyms in the abstract that are not defined. I assume they should be defined, but I'm not sure of the journal's policy on this.

Thank you. We agree that undefined acronyms are not desirable. We have inserted definitions.

Page 2, line 35-37:

The order and logic seems reversed here. Wouldn't it make more sense to say that "...

combining observations from multiple missions ... must be addressed". And then "Therefore, ... measurement accuracy needs to be characterized ..."?

We agree with this point. We have modified the text (line 39-40).

Page 4, line 125:

Would it be more correct to say "A benefit of the single-frequency data set is that we can assess the impact on the dual-frequency data set ..."?  Was the CDAAC processing different (e.g., different filtering methods) in AS-on periods than in AS-off periods?

We appreciate this comment. We have modified the text on Line 133 in the updated manuscript.

Page 5, line 153:

"... are of equal but opposite sign". I suppose it should be 'size' after 'equal'.

We have clarified the meaning of this sentence (line 163).

Equation 1:

What is actually meant by 'delay' here? Is L (disregarding noise) the same as the optical path length minus the distance between transmitter and receiver?

Yes. We have clarified the meaning on Line 168 of the modified manuscript and cleared up some notational issues regarding what is referred to ionospheric delay.

Equation 3:

I think the signs on mu_1 and nu_1 should be opposite of what they are in this equation. Otherwise the math does not work out.

Thank you for catching this error. We have corrected it.

Page 6, line 170:

Why not just add (1) and (2) and divide by two? Wouldn't that give the same without the intermediate calculation of I? Or is there another reason for the intermediate calculation? I think it becomes clear later why, but that should perhaps be mentioned already here.

The purpose of equation (3) is to show how I is calculated from (1) and (2). Equation (3) shows how the ionospheric contribution is calculated from the range and phase measurements.

Adding (1) and (2) eliminates I, so it may be less clear as to how I is calculated. We have not modified the equations.

Page 6, line 175:

I do not understand this argument about the bandwidth. I am fine with that there in principle is no raypath separation at all between L1 and P1 measurements. These come from the same signal, and in the geometrical optics formulation at L-band frequencies there is only one ray path for that signal. Mathematically, L1 phase and P1 pseudorange relate to the integral of the refractive index and group refractive index, respectively, along this path. However, it is unclear to me if this avoids raypath-induced residual ionospheric errors in the further processing. When using eq.(3) to remove the ionospheric delay from L1 in eq.(1), you still have the integration along the original L1 path embedded in the eta term (as I understand the equations and the notion that this is a calibrated GPS signal, eta is the integral along the L1 path of the neutral atmospheric part of the refractive index minus the distance between transmitter and receiver). Thus, a part of eta depends on the ionospheric gradients because the path of the L1 signal depends on the ionospheric gradients. I don't think that is negligible in the context of residual ionospheric errors. How do you deal with this in the further processing? What are the equations that you use? I think these equations needs to be given, so that the results can be reproduced by others, and so that it can be understood to which degree raypath-induced residual ionospheric errors are present.

We appreciate that the reviewer has brought up these points. There are two main comments here: 1) bandwidth and 2) the eta term.

Regarding the bandwidth, our point is that the P-code signal can be decomposed into a signal of multiple frequencies, and is not characterized by a single frequency. The frequencies comprising the P-code signal span 10 MHz. Each frequency in this signal will travel a slightly different path, but this effect is negligible in comparison to raypath separation between L1 and L2. We have clarified the language near Line 188.

The comments regarding the eta term are well founded, but do not contribute to residual ionospheric error in the sense referred to in this paper. The ionosphere affects eta because bending in the ionosphere will slightly shift the tangent height of the raypath in the neutral atmosphere. Since ionospheric bending is quite small, this shift is similarly small and generally ignored in radio occultation processing, for either single or dual frequency processing. (See Hajj et al., 2002 for a processing description).

Page 7, line 191-195:

Here it becomes clear that the sampling rate is not the same for L and P. Is smoothing of I in eq.(3) and/or interpolation there not necessary? It is mentioned that smoothing is applied directly to $P_1$. Is that really correct? Don't you have atmospheric variation in $P_1$ that you don't want to smooth (eta in eq.(2))? Why not apply the smoothing to I in eq.(3) where the

atmospheric variation has been removed? Well, I think it is when I read on. Revision in the text here is probably needed.

Thank you for this comment. We agree this section was not quite expressed correctly. We have made clarifications on Lines 220-224.

Page 8, line 225-226:

Is it correct to use the word bias here? If it is different from profile to profile (with different noise), it is not a systematic error, and thus not a bias. The approach seems good, although I am not an expert in filters and smoothing. There will of course be a residual error after the smoothing no matter how you do it, and there will be error correlations between adjacent points, but I don't think you can call that a bias.

We appreciate this comment. We agree that the term "bias" leads to confusion here. The use of the term bias here is not the same as we refer to elsewhere (e.g. climatological bias or systematic error as used in the reviewer's comment). We have introduced the additional modifier "statistical" (near line 256) to characterize the bias and included a reference where statistical biases due to weighted averages are discussed.

Page 8, line 234-235:

Here it is explained that the smoothing is applied to the ionospheric estimate in (3). I think that contradicts the information on page 7, where it was stated that smoothing is applied directly to $P\_1$. It makes more sense if it is applied to I at 1-second intervals in (3). But I think the text on page 7 needs to be revised to be consistent with this.

We thank the reviewer for bringing a manuscript error to our attention. The smoothing is applied to the ionospheric estimate I (Eq. 3), not to P1. I is available at 1-sec cadence. We have corrected the manuscript (line 220).

Page 8, line 247:

I think I understand the approach, but I don't understand the sentence here. What is 'formal variance'?

We have clarified the sentence and added a reference (lines 290-294).

Page 9, line 262:

This is basically the same as just mentioned two lines before.

We have removed the repetition in the preceding lines (line 304).

Page 9, line 274-275:

Is it really smoothing of L2, or is it rather smoothing of L1-L2? In any case it is not the frequency that is smoothed, but the phase. And not the lower SNR that is mitigated, but rather the negative effects of it. The language in the paper could be more precise here.

We have modified the language to be more precise (line 319).

Page 9, line 315-317:

The sentence here does not really make sense to me. Could it be rephrased?

We believe sufficient explanation is made in the preceding sentences of the paragraph, so we have removed this sentence.

Page 11, line 332:

Could you provide one or more references to support "Other climate-related work" here?

References are now added at the end of the sentence (line 387).

Figure 4:

The figure shows median values. Would it be similar with mean values? Or are there a number of outliers that makes the median and mean significantly different?

Yes, due to outliers the mean and median can differ significantly, particularly for the small differences found in these comparisons. We believe that median is an acceptable metric for the comparison as it is a robust statistic.

Page 12, line 365:

Common profiles in three data sets? I suppose there are not common profiles between the CDAAC AS-on and CDAAC AS-off data sets. Either AS was on or it was off. It cannot be both at the same time for the same profiles. So should it rather be common profiles in two datasets here?

Thank you for this comment. We now refer to two pairs of profile sets and provide more clarity on why the differences can occur (near line 423).

Page 12, line 366-369:

Could the reason for the differences also be that these are median values? With mean values one would expect to be able to see consistency between differences in Figure 4 and Figure 6,

since this is more or less linear algebra. But with median values it is a more complex and different story, and one cannot generally expect such consistency. It is difficult to see the differences in altitudes without a grid in the figures, but they seem quite similar (I do see a small offset at 10 km).

Thank you for this comment. This is a valid point. Even in the case of the mean, full consistency would not occur because differences are formed between the JPL and CDAAC data sets at altitudes that differ from the differences formed between the CDAAC and MERRA-2 data sets. We have changed the language to be more explicit and also suggest the point made about median differences (lines 425-434).

Page 13, line 390:

ECMWF-Interim? Do you mean ERA-Interim?

Corrected.

Page 14, line 449-450:

I was not able to find this estimate (1% near 30 km) in (Danzer et al., 2013). The number seems at least an order of magnitude too large. Danzer et al. (2013) show mostly errors/differences in bending angle, but at much higher altitudes. They show also the effect on temperature profiles (in their Fig. 8), but biases in temperature at 30 km can be very different from biases in refractivity, because of downward error propagation via the hydrostatic integration and large biases in the retrieved pressure. In the introduction of Danzer et al. (2013), they cite error estimates in previous works (Schreiner et al., 2011) of 0.045% at 30 km for refractivity, which sounds much more reasonable to me.

A couple of questions related to this: What would be the size of errors in bending angle (in micro-radians), that typical pseudorange noise could create in single-frequency processing? And how would this compare to expected residual ionospheric errors in dual-frequency processing? I think you would be able to answer these questions with the data that you have: You could take the derivatives with respect to impact altitude (in m) of the differences between the L1-L2 and CA-L1 fits in figure 3 (examples c, d, e, and f). That should give you four examples of bending angle errors (in radians) due to pseudorange noise between 15 and 60 km. I don't know the answer myself, but me feeling is that it will be difficult (even when averaging over many profiles) to say anything conclusive about the residual ionospheric errors using the single-frequency processing because of the pseudorange errors. In any case, it would be very interesting and very relevant to see example estimates (and perhaps also ensamble averages) of the bending angle errors from the single-frequency processing with this straight-forward approach. I strongly suggest such estimates to be included in the paper.

**Citation**: https://doi.org/10.5194/amt-2021-241-RC2

Thank you for this comment. We agree that an error was made in the paper. We have now corrected the error and added a figure and paragraph related to bending angle precision by comparison with JPL dual frequency processing for AS-off periods. See lines 538-558.

---

## Author Response (AR2)

I find it confusing to use the word 'delay' in the text around Eq. (1). In principle 'delay' should refer to a time interval, but that is not the way it is used here. When talking about a phase measured in meters, it would make more sense to use the word 'phase path'. I think 'group delay' (as in eq. 2) could be replaced by 'pseudorange'.

We have reworked the manuscript and removed the word delay in several places. We now refer to pseudorange and phase. The word "delay" is now used exclusively in reference to pseudorange where the word "delay" has an appropriate meaning. For pseudorange, the slower group velocity in the ionospheric medium causes a delay in the signal relative to propagation in free space. We have also removed the noise terms in Equations (1)-(3) since they are no longer needed.

Line 177-180: Raypath separation between P1 and L1 signals does not make sense to me. To my understanding (and I'm rather sure of this), P1 and L1 measurements are related to the same path. Because of the P-code (which is modulated on the phase) the wave can be decomposed into multiple waves with slightly different frequencies. The ray path (in geometric optics) of such a wave is perpendicular to the wavefront of the carrier, even when the signal is comprised of multiple signals with slightly different frequencies around the carrier frequency. I think the authors should remove line 177-180 or provide a reference to a text book to support their statement.

We have removed the statement and also the statements closer to lines 500-510 in the manuscript.

On the related issue about eta in Eqs. (1) and (2): In their response, the authors agree with me, and correctly points out that the ionosphere affects eta because bending in the ionosphere will slightly shift the tangent height of the ray path in the neutral atmosphere, but then goes on to say that it is quite small and is ignored in radio occultation processing. However, my comment was that I don't think it is negligible in the context of residual ionospheric errors/bias. As I understand it, ray path separation is less of a problem in bending angle correction, but in phase correction (as here) it is a larger problem. I disagree that the residual ionospheric error, regardless of whether we say it is due to ray path separation or tangent height shift (it is basically the same thing), is avoided with single-frequency processing. The discussions in lines 459-465, 471-474, 482-486, 500-505, and 543-547, that single-frequency processing is not (or very little) affected by residual ionospheric bias cannot stand without mathematical evidence to support such a claim. To my knowledge, no such evidence exists, and I think the authors should remove the indications that the single-frequency processing (no matter how many profiles you average over) can resolve the residual ionopsheric bias in dual-frequency processing.

We appreciate this comment of the reviewer. It is useful to understand the residual ionospheric error in the context of single frequency processing. A great deal has been written on the topic of RIE, which we now present in an Appendix. The analysis shows that under many conditions such as near solar maximum daytime, the dual-frequency RIE can significantly exceed single-frequency RIE. Text has been modified and an Appendix added to the manuscript. The modified discussion is near Line 545 of the Track Changes version.

Something is not right with the added sentence in line 194-195: "CDAAC processing uses the BERNESE software for RO orbit and clock determination, and for retrievals, are described in Schreiner et al. (2009) and Kuo et al. (2004), respectively."

We have corrected the language near Line 207.

I was not able to find anything about statistical bias in Section 7.2 of Mandel, 1964. Or in any other section. Anyhow, I think the description with weighted averaging seems sound, and I wonder if the notion about a possible 'statistical bias' is necessary in this context.

We no longer refer to "statistical bias" but continue to refer to unbiased estimators as referred to in the work by Mandel. Near Line 313.

Figure 8 caption: "The blue curve" is black in my copy.

We have made the correction.

Sean Healy, Andras Horányi are no longer co-authors? But they are mentioned in the Author Contribution: S. Healy and A. Horányi provided review and editing of the manuscript and elements of the conceptualization and methodology.

The co-authors Healy and Horányi felt their contributions did not merit co-authorship, so we have removed them as co-authors and we now mention Healy in the acknowledgements. Horányi is no longer mentioned because he worked with Healy, and it would seem inappropriate for Healy to acknowledge Horányi's contribution in the Acknowledgements since Healy is no longer a co-author.

---

## Author Response (AR3)

Report #1

Submitted on 28 Mar 2022
Anonymous referee #2

**Suggestions for revision or reasons for rejection (will be published if the paper is accepted for final publication)**

I am happy with the revisions, except one major issue: In response to my previous comment about eta in Eqs. (1) and (2), the authors have now included a new Appendix where they compare higher-order residual terms associated with single- and dual-frequency processing. In their answer they claim that the analysis shows that the dual-frequency residual ionospheric error (RIE) can significantly exceed single-frequency RIE. The appendix is very nicely written, but it does not show what the authors claim it shows. The equations in the Appendix is only about the higher order terms in the Appleton-Hartree formula, there are no equations for the raypath separation effect. In line 459 in the main text, the authors write that the dominant raypath separation effect is absent in the single-frequency processing. I agree that it is the dominant effect, but there is no proof in the Appendix that it is absent in the single-frequency processing. The authors refer to studies by Hardy et al. (1994), by Syndergaard (2000), and by Hoque and Jakowski (2010). As noted in the Appendix, the latter study finds that a bending term (or curvature correction term in Hardy et al (1994), or dispersion residual in Syndergaard (2000)) exists on both L1 and L2 and far exceeds the second order residual. I am pretty certain that this term (in different variants in the three papers) does not disappear in the single-frequency processing. In single-frequency processing only the L1 path is used, but the effect is still there since the L1 path is affected by the ionosphere, and the extra bending is contained in eta (see my earlier comments about eta in Eqs. (1) and (2)). In the dual-frequency processing, one can call it raypath separation effect, but in the single-frequency processing 'bending term' is probably a better name for it. I believe the frequency dependency in the single-frequency processing (residual after ionospheric correction) will be as $1/f1^4$, whereas in the dual-frequency processing I believe it is $1/(f1*f2)^2$. In both cases, the factor in front of these is so large that the bending term dominates the much smaller higher-order residual terms. It is worth noting that we are talking about single- and dual-frequency excess phase corrections here, and that the dominating term is greatly reduced in the dual-frequency bending angle correction.

I think the authors should either include the bending term in the new Appendix, and give relative estimates of its size in single- and dual-frequency excess phase processing (as nicely done with the higher order residuals), or alternatively the Appendix could be removed again if the authors disagree that this term is present also in the single-frequency processing. In any case I think the authors should remove the indications in lines 454-460, 469-473, 481-485, and 535-539 of the revised manuscript, as well as in the Appendix that the single-frequency processing has the potential to resolve the residual ionospheric bias in dual-frequency processing.

Response: we appreciate this comment from the reviewer and have modified the manuscript accordingly. The impacts due to raypath bending on the single frequency correction are now described in the appendix. The possible benefits of comparing the dual- and single-frequency

correction methods now take this into account. We have accordingly modified the manuscript in the lines mentioned by the reviewer above.